# High-speed trains versus air transport vectors for mass transfers of critically ill patients: The TRANSCOV cohort study

Chatpimuk Thipayamaskomon[1], Olivier Grimaud[1]*, Pierre Tattevin[2], Lionel Lamhaut[3], Emmanuelle Leray[1], Noemie Letellier[4], Sahar Bayat[1], Christophe Fermanian[1], Sylvie Martin[1], Jean-Marc Philippe[5], Eric Maury[6], Marc Noizet[7], François Braun[8], Manuel Dolz[9], Marc-Antoine Sanchez[10], Hélène Coignard-Biehler[11], Nathalie Prieto[12], Hugues Delamare[13], Virginie Cayré[14], Pierre Carli[3], Albert Vuagnat[15], Julien Pottecher[16], Agnès Ricard-Hibon[17], the TRANSCOV Investigators

1 Univ Rennes, EHESP, CNRS, ARENES – UMR6051, Rennes, France, 2 Pontchaillou University Hospital, Rennes, France, 3 Department of Anaesthesiology and Intensive Care, SAMU de Paris, Hôpital Necker, Paris, France, 4 Univ Rennes, Inserm, EHESP, Irset (Institut de recherche en santé, environnement et travail), UMR_S1085, Rennes, France, 5 General Directorate for Health (DGS), French Health and Social Affairs Ministry, Paris, France, 6 Université Pierre et Marie Curie Faculté de Médecine, Paris, France, 7 Department of Infectious Diseases and Intensive Care Unit, Emergency Department and Service d'Aide Médicale Urgente (SAMU) 68, Groupe Hospitalier de la Région Mulhouse Sud Alsace, Mulhouse, France, 8 CHR Metz-Thionville, Metz, France, 9 Direction centrale du service de santé des armées, Paris, France, 10 Information Systems and Digital Department (DSIN), French Army Health Service, Saint Mandé-Bat14, 69 avenue de Paris, Saint-Mandé, France, 11 Emergency Medical Service, Lyon University Hospital, Lyon, France, 12 Cellule d'Urgence Médico-Psychologique (CUMP), Centre Régional du Psychotraumatisme (CRP), Hôpital Edouard Herriot, HCL, Lyon, Auvergne-RhôneAlpes, France, 13 Santé publique France, The French Public Health Agency, Saint-Maurice, France, 14 Agence régionale de santé (ARS) Grand Est, Nancy, France, 15 Department for Research, Studies, Evaluation and Statistics (DREES), French Health and Social Affairs Ministry, Paris, France, 16 Anaesthesiology, Critical Care and Perioperative Medicine, Hôpitaux Universitaires de Strasbourg, Strasbourg, France, 17 Emergency Department SAMU-SMUR 95, CHG Pontoise-Beaumont/Oise, Pontoise, France

¶ Membership of the the TRANSCOV Investigators Group is listed in the Acknowledgments.
* Olivier.Grimaud@ehesp.fr

## Abstract

### Background

The first COVID-19 epidemic wave hit the East and Ile-de-France regions in France, resulting in overwhelmed intensive care units (ICUs). Alongside helicopters and planes, high-speed trains were used for the first time to mass-evacuate critically ill patients. This study aimed to compare outcomes of patients evacuated by trains and by aircrafts.

### Methods

This was a multicentre retrospective cohort study. Between 13 March and 10 April 2020, 38 ICUs in France transferred patients with severe COVID-19 to 60 ICUs in unaffected regions and countries. Patients were divided into the train group (n = 130) and the air group (n = 163). The study outcomes included 28-day case-fatality,

**Data availability statement:** In accordance with current regulations, data from the TRANSCOV database cannot be made public. The database is hosted at EHESP. Under the GDPR, the Director of EHESP acts as the data controller. To gain access to this data, a request must be submitted to the EHESP Data Protection Officer (contact: dpo@ehesp. fr). Access will also be subject to the approval of the Transcov Scientific Committee and authorisation from the French Data Protection Authority (www.cnil.fr).

**Funding:** The study was funded by the Directorate General of Health (DGS) of the French Ministry of Health (https://sante.gouv.fr) and received supplementary funding from the French ANRS for Emerging Infectious Diseases (https://anrs.fr/en/homepage). CT, OG, CF and SM received the funds. The funders had no role in the study design, data collection and analysis, decision to publish, or preparation of the manuscript.

**Competing interests:** The authors have declared that no competing interests exist.

destination ICU length of stay and post-transfer Sequential Organ Failure Assessment (SOFA) score.

## Results

Age and comorbidity did not differ between groups. Although patients spent more time (+2 hours) and travelled further (+250 km) in the train group than in the air group, the median post-transfer SOFA score was lower in the train group (6 vs 7; $p = 0.03$). The 28-day mortality rates were not different (train/air unadjusted incidence risk ratio: 0.96; $p = 0.94$). The ICU stay duration was shorter (−6 days) in the train group, but this difference was reduced after adjusting for clinical events, such as nosocomial infections.

## Conclusion

High-speed train was a safe vehicle for remote transfer of critically ill patients. The selection of healthier patients and the better physiological and care conditions during the evacuation may explain the shorter ICU stays of patients transferred by trains.

## Introduction

During the first Coronavirus disease 2019 (COVID-19) epidemic wave, intensive care units (ICUs) in the East and Ile-de-France regions of France were overwhelmed by a surge of critically ill patients. To relieve the pressure, 661 medical evacuations were organized from 70 overcrowded "origin" ICUs to 146 "destination" ICUs located in less affected French regions and neighbouring countries between 13 March and 10 April 2020.

Emergency services, fire brigades, civil security and army worked together with the health authorities to carry out this unprecedented operation [1]. Ten high-speed trains (TGVs in French) were used for the first time as the main transport vector to evacuate 202 patients to distances ranging from 350 to 900 km. In less than 24 hours, a high-speed train with eight double-decker carriages could be organized. Six patients were transported and accompanied by two doctors, four nurses and a logistician per carriage – four carriages per train were used for patient care. In addition to cardiorespiratory monitoring and ventilation equipment (available on all modes of transport), each train convoy was equipped with a portable laboratory and an echocardiograph. However, these two items of equipment were not available in helicopters, which were used for around 70% of air transfers during this period. More details on the train preparation and management are available elsewhere [2–3].

Regardless of the transport vectors used, patients selected for medical evacuation were clinically stable and met the following criteria: confirmed COVID-19 diagnosis, weight <100 kg to facilitate transportation, mechanical ventilation for at least 24 hours, no renal replacement therapy, fraction of inspired oxygen <60%, positive end-expiratory pressure ≤14 cmH$_2$O, noradrenaline dose <1 mg/h, no prone position

in the last 24 hours, and consents from the relatives [1]. Previous analyses showed that transferred patients did not experience major clinical events during the evacuation [4–7].

In this article, we compared initial profiles and clinical consequences of the patients transferred by trains compared to air transport vectors.

## Methods

### Design

The TRANSCOV cohort is a multicentre observational retrospective study to evaluate the impact of long-distance ICU-to-ICU mass transfers organized from two French regions (East Region and Ile-de-France) toward other regions or neighbouring countries between 13 March and 10 April 2020. Full details of the cohort are available elsewhere [8].

### Data

Research assistants filled in electronic case report forms (CRFs) by extracting information from the medical records in the "origin ICUs" (where patients were initially admitted) and "destination ICUs" (where they were transferred to). The "main transport vector" (i.e., the means of transportation used for the longer part of the journey) was the exposure variable. It was either identified in a dedicated "CRF_transfer", or if missing, inferred by matching records from the TRANSCOV database with the logs of medical evacuations from the French Ministry of Health (matching variables: patient month and year of birth, date of transfer, origin and destination towns). For several records, the logs indicated main transport vectors as "air", without specifying whether it was a plane or helicopter. To keep the corresponding patients in the analysis, all patients transferred by planes or helicopters were grouped into "air" category. The few patients whose main vectors were land ambulances and boats were excluded.

The total transfer duration was deduced from the departure time at the origin ICU and the arrival time at the destination ICU. The distance travelled in the main transport vector was calculated using Google Maps. This distance was the length of a straight line between airports or between city centres for planes and helicopters, respectively. For train transfers, it was the shortest railway route between the origin and destination train stations. Google Maps was also used to estimate the travel time of plane and train transfers. For helicopter transfers, the average helicopter speed for transferring critically ill patients reported by Hong et al was used [9].

For each transferred patient, the TRANSCOV database contains age, sex, height, weight, comorbidities, functional status (Knaus score) before ICU admission, clinical severity based on the lung damage extent and the Simplified Acute Physiology Score II (SAPS II). The database also included information on significant clinical events that occurred during the ICU stay, such as shock, acute kidney injury (AKI) and nosocomial infections (yes/no variables). It also contained details on treatments, including mechanical ventilation, neuromuscular blockade, prone position, tracheostomy and extracorporeal membrane oxygenation.

The main outcomes were post-transfer 28-day ICU mortality and duration of destination ICU stay (hereafter, referred to as length of stay). The transfer immediate impact was estimated by comparing a set of parameters measured at the origin ICU just before the transfer and at the destination ICU upon arrival. These parameters included catecholamine requirement, the arterial oxygen partial pressure to fraction of inspired oxygen ($PaO_2/FiO_2$) ratio, blood gas parameters and the Sequential Organ Failure Assessment (SOFA) score.

### Statistical analysis

Participants were described according to train and air groups using medians and interquartile ranges for continuous variables and numbers and percentages for categorical variables. The Mann-Whitney's and chi-square tests were used to identify significant differences between groups. Linear regression and modified Poisson regression models were used

to estimate associations between exposures and outcomes, including length of stay (log transformed to account for the non-normal distribution) and 28-day mortality at the destination ICU. As mixed effect models did not suggest within-centre correlations (origin or destination ICU), a simple regression analysis was used for both outcomes.

Univariable associations were investigated between each outcome and explanatory variables, categorized as "initial patients characteristics", "transfer characteristics" and "treatment and events at the destination ICU". Multivariable models were used to examine how these categories influenced the associations between the main transport vector and outcomes. Only variables with $p < 0.25$ in the univariable analysis were entered in the multivariable models. The final model was obtained by backward stepwise elimination.

Mediation analysis was used to determine whether clinical events contributed to the association between transport vector and length of stay [10]. Sub-group analyses were performed to evaluate the consistency of the findings using age (terciles), clinical status before transfer (below vs above the median SAPS II at admission at the origin ICU), SOFA score before and after transfer (below vs above the median value), period of transfer (weeks 1–2, week 3, and week 4 of the study period), region of origin (East and Ile-de-France) and region of destination (grouped as Northwest vs Southwest). Sensitivity analysis involved a subsample of patients whose main transport vector was recorded (as opposed to inferred).

Analyses were performed with SAS version 9.4 (SAS Institute Inc) for data management and description and with R version 4.4.1 for model running. Statistical significance was defined as a two-sided $p < 0.05$. This article follows the STrengthening the Reporting of OBservational studies in Epidemiology (STROBE) guidelines.

## Ethics

The TRANSCOV cohort was approved by the independent Ethical and Scientific Committee (CESREES) on 16 July 2020 (file no. 2046524). CESREES waived the need for oral or written consent. However, origin hospitals sent written notices to the patients or their legal guardians that explained the objectives and methods of the study, as well as the right and procedures to decline participation. Data were accessed from 1 February until 31 July 2024. Data were anonymised at the data collection stage. Authors had no access to identifying information.

## Results

In total, 44 of the 70 origin ICUs (63%) took part in the analysis and provided information on 501 transferred patients. Participation was lower among destination ICUs (65/146, 45%). Consequently, the destination data were missing for 171 patients. Most of these patients (125/171, 73%) were evacuated to ICUs located in neighbouring countries where ethical agreements could not be obtained. The main transport vector could not be determined for 20 patients. The few patients evacuated by boat (n = 6) and land ambulance (n = 11) were excluded (Fig 1). Therefore, the sample included 293 patients distributed between the train group (n = 130) and the air group (n = 163). Considering the French territory, this corresponded to an estimated participation rate of 64% (130/202) in the train group and 57% (154/269) in the air group. The routes taken by patients involved in the analysis are shown in the supplement (S1 Fig). Baseline characteristics of patients excluded from the analysis were not different from those of the included patients, except for higher weight (+4 kg) and lower rate of chronic alcoholism (0.5% vs 4.4%) (S2 Table). The main transport vector was recorded for 193 patients and inferred for 100 patients. The baseline characteristics were not different in the reported and inferred groups, except for higher lung damage extent in the recorded group (lung lesions >50%: 53.3% vs 35.4%; $p = 0.04$) (S3 Table).

Age, weight and comorbidity profiles were similar in both groups (Table 1). At admission in the origin ICU, severity was lower (−4 SAPS II points; $p = 0.053$) and intubation was more frequent in the air group (52.5% vs 37.7%). Patients who were evacuated by trains stayed longer in the origin ICUs before transfer (median: 6 vs 5 days; $p = 0.02$).

Train transfers were organized mostly to the Northwest and Southwest regions of France and toward the end of the study period. Typically, patients spent an estimated 3h22min in a train (vs 1h25min in an air transport vector) and travelled ~600 km (vs 350 km by air) (Table 2). The total transfer duration (ICU door to door) by train was nearly twice as long as by

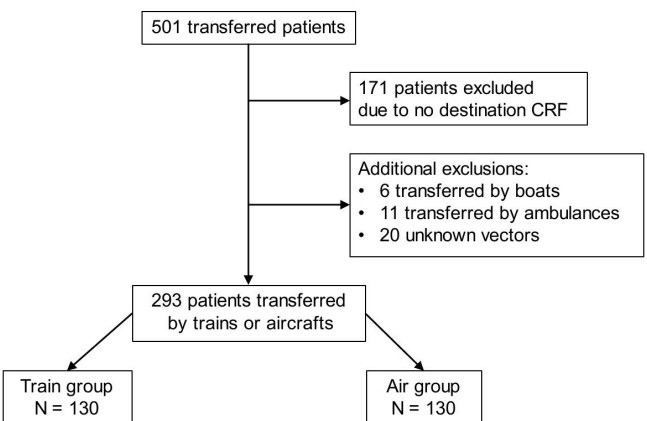

**Fig 1. Study flowchart of participants.** CRF, case report form.

**Table 1. Comparison of demographic data, clinical profile and clinical events before transfer by trains or aircrafts.**

| Patient characteristics | Train group N=130 | Air group N=163 | P-value[a] |
|---|---|---|---|
| Age (years) | 66 (57–71) | 63 (56–70) | 0.19 |
| Men | 99/130 (76.2) | 114/163 (69.9) | 0.24 |
| Weight (kg) | 80 (73–90) [6] | 82 (74–94) [11] | 0.07 |
| Risk factors | | | |
| Current smoking | 5/121 (4.1) | 7/154 (4.5) | 0.87 |
| Chronic alcoholism | 5/119 (4.2) | 7/152 (4.6) | 0.87 |
| Comorbidities | | | |
| Any comorbidity | 91/130 (70.0) | 108/163 (66.3) | 0.50 |
| No. of comorbidities | 1 (0–2) [3] | 1 (0–2) [6] | 0.80 |
| Diabetes | 38/130 (29.2) | 52/162 (32.1) | 0.60 |
| Hypertension | 70/130 (53.8) | 81/162 (50.0) | 0.51 |
| Cardiovascular disease[b] | 16/128 (12.5) | 18/160 (11.3) | 0.74 |
| Asthma or COPD | 9/127 (7.1) | 21/161 (13.0) | 0.10 |
| Functional limitations (KNAUS score) | 49/117 (41.9) | 55/150 (36.7) | 0.39 |
| Clinical status at admission in the origin ICU | | | |
| Disease duration before ICU admission (days) | 8.0 (6.0–12.0) [7] | 9.0 (6.0–11.5) [3] | 0.61 |
| SAPS II | 42 (34–50) [8] | 38 (29–49) [27] | 0.05 |
| Intubated at admission | 49/130 (37.7) | 85/162 (52.5) | 0.01 |
| LOS at the origin ICU (days) | 6 (3–10) [1] | 5 (3–8) [1] | 0.02 |

Data are reported as median (IQR) [n missing] or n/total n (%). COPD, chronic obstructive pulmonary disease; CT, computed tomography; ICU, intensive care unit; IQR, interquartile range; LOS, length of stay; SAPS, Simplified Acute Physiology Score.

a P-values were calculated with chi-square test or Fisher exact test for discrete variables and with Wilcoxon-Mann-Whitney test for continuous variables.

b Cardiovascular disease included either ischaemic heart disease, heart failure or stroke.

air. Comparison of the pre- and post-transfer clinical status suggested a worsening of physiological parameters after medical evacuation. Specifically, the $PaO_2/FiO_2$ ratio diminished from 175.8 to 141.8 mmHg in the air group (p<10−3) and from 182.9 to 160.8 mmHg in the train group (p=0.10). However high percentage of missing data for pre-transfer SOFA score

**Table 2. Transfer characteristics and patient clinical status before and after the transfer.**

| Variables | Train group N = 130 | Air group N = 163 | P-value[a] |
|---|---|---|---|
| Transfer characteristics | | | |
| Period of transfer | | | <0.001 |
| 13–26 March 2020 | 18/130 (13.8) | 26/163 (16.0) | |
| 27 March to 2 April 2020 | 47/130 (36.2) | 102/163 (62.6) | |
| 3–10 April 2020 | 65/130 (50.0) | 35/163 (21.5) | |
| Origin region | | | <0.001 |
| Bourgogne Franche Comté | 0/130 (0.0) | 27/163 (16.6) | |
| Grand Est | 50/130 (38.5) | 53/163 (32.5) | |
| Ile de France | 80/130 (61.5) | 83/163 (50.9) | |
| Destination region | | | <0.001 |
| Abroad | 0/130 (0.0) | 9/163 (5.5) | |
| Northwest region | 69/130 (53.1) | 60/163 (36.8) | |
| Southwest region | 61/130 (46.9) | 33/163 (20.2) | |
| Southeast and central regions | 0/130 (0.0) | 61/163 (37.4) | |
| Distance (km) | 590 (466–835) | 344 (227–496) | <0.001 |
| Duration (hours) | | | |
| ICU door to door | 8.6 (6.9–10.1) [16] | 4.8 (3.0–6.0) [25] | <0.001 |
| In the main transport vector | 3.4 (2.2–4.4) | 1.4 (1.1–1.9) | <0.001 |
| Pre- and post-transfer clinical status | | | |
| SOFA score | | | |
| Pre-transfer | 6.0 (4.0–7.0) [62] | 5.0 (3.0–7.0) [60] | 0.59 |
| Post-transfer | 6.0 (3.0–7.5) [22] | 7.0 (4.0–10.0) [30] | 0.03 |
| $PaO_2/FiO_2$ ratio (mmHg) | | | |
| Pre-transfer | 182.9 (154.0–224.0) [5] | 175.8 (148.3–230.0) [5] | 0.45 |
| Post-transfer | 160.8 (118.8–202.3) [2] | 141.8 (106.2–195.0) [0] | 0.07 |
| Arterial pH | | | |
| Pre-transfer | 7.44 (7.40–7.47) [4] | 7.41 (7.37–7.44) [1] | <0.001 |
| Post-transfer | 7.41 (7.35–7.46) [1] | 7.40 (7.34–7.44) [0] | 0.12 |
| $PaCO_2$ (mmHg) | | | |
| Pre-transfer | 41.5 (38.0–46.0) [4] | 44.0 (38.6–50.1) [1] | 0.02 |
| Post-transfer | 44.0 (40.0–50.0) [1] | 45.0 (39.4–53.0) [0] | 0.59 |
| Serum bicarbonate (mEq/L) | | | |
| Pre-transfer | 28.0 (25.3–30.0) [7] | 26.8 (24.0–30.0) [2] | 0.17 |
| Post-transfer | 28.4 (25.5–31.2) [3] | 27.4 (24.6–30.8) [1] | 0.16 |
| Catecholamine requirement | | | |
| Pre-transfer | 42/129 (32.6) | 51/162 (31.5) | 0.84 |
| Post-transfer | 49/128 (38.3) | 61/163 (37.4) | 0.88 |
| Noradrenaline dose (mcg/kg/min) | | | |
| Pre-transfer | 0.1 (0.1–0.2) [4] | 0.1 (0.1–0.2) [3] | 0.34 |
| Post-transfer | 0.2 (0.1–0.2) [4] | 0.2 (0.1–0.2) [8] | 0.15 |
| Post-transfer SAPS II | 39 (32–49) [11] | 40 (30–51) [10] | 0.69 |
| Ventilation at admission in the destination ICU | | | 0.67 |
| Naso or orotracheal tube | 126/130 (96.9) | 160/163 (98.2) | |
| Tracheostomy | 3/130 (2.3) | 3/163 (1.8) | |
| No ventilation | 1/130 (0.8) | 0/163 (0.0) | |

Data are reported as median (IQR) [n missing] or n/total n (%). FiO2, fraction of inspired oxygen; ICU, intensive care unit; IQR, interquartile range; PaCO2, arterial carbon dioxide partial pressure; PaO2, arterial oxygen partial pressure; SAPS, Simplified Acute Physiology Score; SOFA, Sequential Organ Failure Assessment.

[a]P-values were calculated with chi-square test or Fisher exact test for discrete variables and with Wilcoxon-Mann-Whitney test for continuous variables.

(62/130 (48%) and 60/163 (37%) in the train and air groups respectively) made comparisons more difficult. Nevertheless, on the basis of available data, the pre-transfer SOFA scores were similar between groups, whereas the post-transfer SOFA scores were higher in the air group than the train group (median post-transfer SOFA scores: 7 vs 6; $p = 0.03$).

One or more clinical events during the stay at the destination ICU were reported for ~80% of patients in both groups (Table 3). Except for encephalitis and epilepsy (<2% in both groups), the frequency of specific adverse events was lower in the train group, particularly AKI (train 19.2% vs air 32.5%; $p = 0.01$) and nosocomial infections (51.5% vs 65.6%; $p = 0.01$). Patients transferred by train also required less prone position (34.6% vs 50.3%; $p = 0.007$) and tracheostomy (11.5% vs 26.4%; $p = 0.002$) but had similar mechanical ventilation durations. The 28-day mortality at the destination ICU was low and similar in both groups. However, the length of stay at the destination ICU was shorter in the train group (−6 days, $p = 0.003$). This pattern of shorter lengths of stay in the 'train' group applied whatever the tertile of distance travelled during the transfer (difference in median length of stay: −7, −11.5 and −8 days from 1rst to 3rd tertile).

Crude and adjusted models for patients and transfer characteristics confirmed that the transport vector was not associated with the 28-day ICU mortality (Table 4). The incidence rate ratio decreased to 0.48 after adjusting for events/treatments at the destination ICU, but the association did not reach statistical significance. Conversely, both crude and adjusted models suggested a statistically significant association between train transfers and shorter length of stay. As the length of stay was log-transformed in the linear regression model, the exponentiated coefficients can be interpreted as

**Table 3. Clinical events, treatments and outcomes at the destination ICU.**

| Variables | Train group N = 130 | Air group N = 163 | P-value[a] |
|---|---|---|---|
| Clinical events | | | |
| Any clinical event | 101/130 (77.7) | 137/162 (84.6) | 0.13 |
| Shock | 23/130 (17.7) | 40/162 (24.7) | 0.15 |
| Acute kidney injury | 25/130 (19.2) | 53/163 (32.5) | 0.01 |
| Nosocomial infection | 67/130 (51.5) | 107/163 (65.6) | 0.01 |
| Thromboembolic event | 27/130 (20.8) | 38/161 (23.6) | 0.56 |
| Encephalitis | 2/130 (1.5) | 2/162 (1.2) | >0.99 |
| Epilepsy | 1/130 (0.8) | 1/163 (0.6) | >0.99 |
| Delirium | 28/129 (21.7) | 44/162 (27.2) | 0.28 |
| Other psychological problems | 17/125 (13.6) | 24/158 (15.2) | 0.71 |
| Neuromyopathy | 40/130 (30.8) | 57/160 (35.6) | 0.38 |
| Treatments | | | |
| Neuromuscular blocking agents | 92/128 (71.9) | 122/163 (74.8) | 0.57 |
| Neuromuscular blockage duration (days) | 1.8 (0.8–4.1) [10] | 3.0 (0.9–6.9) [14] | 0.06 |
| Prone position | 45/130 (34.6) | 82/163 (50.3) | 0.007 |
| Tracheostomy | 15/130 (11.5) | 43/163 (26.4) | 0.002 |
| ECMO | 1/130 (0.8) | 6/163 (3.7) | 0.14 |
| Mechanical ventilation duration (days) | 10 (5–17) [8] | 11 (6–22) [20] | 0.10 |
| Outcomes at the destination ICU | | | |
| Length of stay (days) | 14 (9–23) [1] | 20 (12–32) [1] | 0.003 |
| Death in ICU | 8/130 (6.2) | 18/163 (11.0) | 0.14 |
| 28-day mortality | 7/130 (5.4) | 11/163 (6.7) | 0.63 |

Data are reported as median (IQR) [n missing] or n/total n (%). ECMO, extracorporeal membrane oxygenation; ICU, intensive care unit; IQR, interquartile range; SAPS, Simplified Acute Physiology Score.

[a]P-values were calculated with chi-square test or Fisher exact test for discrete variables and with Wilcoxon-Mann-Whitney test for continuous variables.

**Table 4. Associations between main transport vector (train or air) and outcomes at the destination ICU.**

| Models | 28-day ICU mortality | | | ICU length of stay[a] | | |
|---|---|---|---|---|---|---|
| | N train/ N air | IRR (95% CI) | P-value | N train/ N air | Exp(beta) (95% CI) | P-value |
| Crude | 129/162 | 0.96 (0.38–2.46) | 0.94 | 129/162 | 0.79 (0.67–0.93) | 0.006 |
| Adjusted for | | | | | | |
| a) Patient characteristics | 122/135 | 0.85 (0.32–2.29) | 0.75 | 129/162 | 0.78 (0.66–0.92) | 0.003 |
| b) a + Transfer characteristics | 120/135 | 0.88 (0.31–2.45) | 0.80 | 108/132 | 0.79 (0.66–0.94) | 0.01 |
| c) b + Clinical events and treatment at the destination ICU | 98/102[b] | 0.48 (0.21–1.12) | 0.09 | 118/139 | 0.90 (0.81–0.99) | 0.03 |

CI, confidence interval; Exp(beta), exponentiated beta; ICU, intensive care unit; IRR, incidence rate ratio.

a Results are from linear regression models using log-transformed length of stay as dependent variable. Therefore, Exp(beta) can be interpreted as a multiplicative factor. For example, according to the unadjusted model, length of stay was 21% shorter [(1 - 0.79) x 100] for patients transferred by train compared with those transferred by air.

b Because clinical events were recorded as yes/no variables without date of occurrence, only patients with ICU length of stay ≤8 days were included in this model, hence the lower number of participants.

multiplicative factors. Thus, the exponentiated coefficient of 0.79 in the crude model corresponded to a 21% shorter length of stay associated with train transfers compared with air transfers. This difference decreased to 10% after adjustment (see S4 Table for variables included in the fully adjusted model). Given its higher frequency in the air group, nosocomial infection may stand in the causal pathway between air transfer and longer length of stay at the destination ICU. The proportion mediated (i.e., part of the total train/air transfer effect on the length of stay explained by nosocomial infection) was estimated to be ~35%.

Subgroup univariate analyses suggested the association between train transfer and shorter length of stay was clearer for patients with less severe disease (S5 Fig). The transfer week and place (origin or destination ICU) did not modify the main results. In a sensitivity analysis including only patients with recorded main vector (as opposed to inferred), the association between train transfer and shorter length of stay, although not statistically significant, showed similar magnitude (not shown).

## Discussion

In France, the exceptional circumstance during the first COVID-19 epidemic wave led to a pioneering use of high-speed train as the main vector for medical evacuation of ICU patients. To our knowledge, our study is the first to compare the outcome of patients with COVID-19 transferred by trains and air transport vectors. We found both groups had similar initial clinical profiles. Patients in the train group travelled further and spent more time in the transport vector and in transit. The 28-day ICU mortality was low and similar in both groups. However, compared with the train group, the air group stayed 6 more days in the destination ICUs and experienced higher frequencies of nosocomial infections and AKI.

Motivations for inter-ICU transfers vary. During the COVID-19 pandemic, patients transfers were often motivated by the need to manage the sudden rising demand of ICU beds. [2] Most relevant scientific studies focused on transfer preparation recommendations [11,12], transfer process and immediate outcomes [13,14]. Overall, these studies show that the transfer of critically ill, but carefully selected, patients with acute respiratory distress syndrome (ARDS) is safe. For instance, Huq et al found that the physiological deterioration following inter-hospital transfer of 45 critically ill patients with COVID-19 resolved within 24 hours [15]. Less attention has been paid to the vector-specific outcomes. Fagoni et al found that the median ICU stay was longer (15.5 days vs 10 days) for patients transferred by air (n = 65) than by land ambulance

(n = 17), but mortality was similar [16]. A study carried out before the COVID-19 era found, compared with patient transferred by land ambulance (n = 60), that patients with severe sepsis transferred by helicopter (n = 121) experienced significantly more detrimental outcomes (e.g., ARDS) and had longer ICU stay and higher hospital mortality (30% vs 17%) [17].

Our results suggest that train is as safe as aircraft for medical evacuation of critically ill patients with ARDS – possibly even preferable because of its association with shorter stay at the destination ICU. However, this association could be partly or fully explained by the selection bias. As high-speed trains were pioneered as transfer vectors, clinical teams might have selected this transport means for patients with better prognosis than those evacuated by air. Although our data lend little support to this hypothesis, the pre-transfer clinical status might have differed between groups in ways that we could not measure. Longer ICU stays in the air group may also be explained by differences in care environment at the destination ICUs. Travelling further away toward regions less or later affected by the epidemic might have benefited patients in the train group. However, we found no clear relationship between distance travelled during transfer and length of stay in destination ICU. Differences in physical and care environments during the transfer could be another potential explanation. Unlike trains, air transport vectors expose to changes in atmospheric pressure, higher noise levels, vibrations and possibly greater acceleration [18]. Patient management is cumbersome in aircrafts, particularly helicopters. Conversely, trains offered more physical spaces (and more equipments than helicopters) and thus a care environment closer to that of an ICU. Although patients of the train group spent longer time in their main vector, they might had benefited from better nursing care which in turn might have reduced their risk of developing nosocomial respiratory infections.

The TRANSCOV cohort is a unique study purposefully built to evaluate the impact of mass transfers that took place in France in spring 2020. Yet, participation in this cohort was suboptimal because of the lack of resources to involve ICUs in neighbouring countries. However, as no train transfer was organized toward neighbouring countries, patients transferred by air within France were more relevant for comparison. Of note, the database contains data on 64% of patients transferred by trains during the study period. The high proportion of missing data for some key parameters (e.g., post-transfer SOFA score) was another limitation. More information on treatments and clinical events during the stay at the destination ICU (e.g., date of events, nosocomial urinary or respiratory infections) would have been useful to better compare outcomes between groups. In addition, data on the ICU contexts (i.e., patient to staff ratio, epidemic dynamics) in both origin and destination ICUs could have helped to better interpret the results. Also, our study did not address the issue of patient to staff transmission during transport, a risk which might have varied between vectors. Although, to our knowledge, no study has specifically examined this risk, there is no evidence to suggest an increase in contamination, either during this wave of transfers or during subsequent waves. It is worth noting that infection control procedures in both train and air vectors were similar to those followed in ICUs[1].

## Conclusion

Train could be an alternative vector for medical evacuation. A single train can evacuate 24 patients to distant ICUs by mobilizing fewer medical staff members per patient than other transport vectors. Pandemics, natural catastrophes, terrorist attacks or armed conflicts [19] could lead to localized surges of patients requiring ICU care. In such circumstances, high-speed trains appear as a viable option for mass transfer of critically ill patients.

## Supporting information

**S1 Fig. Routes taken by patients involved in the study.**
(DOCX)

**S2 Table. Demographic and clinical profiles at the origin ICU and transfer characteristics of included and excluded patients.**
(DOCX)

**S3 Table. Comparison of patient profile at the origin ICU according to recorded or inferred transport vectors.**
(DOCX)

**S4 Table. Association between co-variables included in the fully adjusted model and length of stay at the destination ICU.**
(DOCX)

**S5 Fig. Associations between exposure and length of stay in subgroup univariate analyses.**
(DOCX)

## Acknowledgments

The TRANSCOV cohort has been labelled as a National Research Priority by the National Orientation Committee for Therapeutic Trials and other research on COVID-19 (CAPNET). The authors thank Yasmine Elgharabawi, Kristell Coat, Mathilde Leonard and Alain Renault from the Centre of Clinical Investigation of Rennes Teaching Hospital for database creation and management, and Elisabetta Andermarcher for revising the English in the manuscript. *Collaborating authors from the TRANSCOV-study group.* Asaël Berger, Laure Abensur Vuillaume, Guillaume Louis, Hélène Beringuer, Sebastien Gette,Thierry Caps, Khaldoun Kuteifan, Christian Fuchs, Lionel Nace, Bruno Mourvillier, Stéphane Gennai,Sonia Foalem, Claudia Ciurel, Cosimo Brigante, Lionel Popoff, Farid Arezki, Anne Weiss, Ferhat Meziani, Francis Schneider, Paul-Michel Mertes, Pascal Lexa, Betty Fleury, Elisabeth Gaertner, Gaëtan Plantefeve, Henri Faure, Guillaume Gelé Decaudin, Yves Cohen, Fréderic Adnet, ElieZogheib, Charles Damoisel, Mathieu Boutonnet, Emmanuel Weiss, Jean-Damien Ricard, Georges-Antoine Capitani, Marie Baron, Frédérique Schortgen, Armand Mekontso Dessap, Eric Lecarpentier, Evelina Ochin, Armelle Severin, Thomas Loeb, Nicholas Heming, Pierre Moine, Djillali Annane, Fabrice Bertrand, Philippe Durand, Lucille Wildenberg, Anatole Harrois, Jean-Louis Teboul, Martial Thyrault, Sabrine Nakaa, Sophie Guilmin-Crepon, Stéphane Dauger, Marie Borel, Matthieu Langlois, Anne-Charlotte Gianinazzi, Franck Verdonk, Bruno Megarbane, Muriel Fartoukh, Lionel Lamhaut, Cédric Bruel, Emmanuel Guérot, Alexy Tran Dinh, Nathalie Zappella, Tiphaine Girard, Philippe Montravers, Jean-Louis Dubost, Marie Grangeon, Gaël Piton, Sébastien Pili-Floury, Thibault Desmettre, Toufiq El Cadi, Bérengère Vivet, Marie Clavier, David Corege, Jean-François Cicala, Paul-Simon Pugliesi, Jean-Paul Denis, Eléonore Timsit, Sébastien Mirek, Mohamed Dyani, René-Gilles Patrigeon, Bernard Lecomte, Laura Federici, Laurent Serpin, Pierre Callige, Clément Dubost, Olivier Richard, Virginie Laurent, Philippe Ichai, Julio Badie, Morlaye Sano, Benoît Plaud, Michael Darmon, Elie Azoulay, Daniel Da Silva, Mourad Lakhdari, Alexandre Boyer, David Tran Van, Camille Foucault, Thierry Mayet, Quentin Levrat, Cédric Darreau, Pierre-Yves Cordier, Raphaël Paris, Marie Soury, Maxime Garnier, Patrick Lafforgue, Romaric Grenot, Arnaud Thille, Pierre Kergoat, Philippe Goutorbe, Fanny Bounes-Vardon, Antoine Galy, Olivier De Soyres, Tahar Saghi, Hubert Grand, Laurent Muller, Hélène Charbonneau, Kevin Quesnel, Jérôme Pillot, François-Michel Beloncle, Pierre Asfar, Sigismond Lasocki, David Schnell, Anne Renault, Olivier Huet, Claude Cohen, Marion Costecalde, Elodie Brunel, Philippe Petua, Audrey Destizons, Caroline Pouplet, Antoine Roquilly, Jean Reignier, Bertrand Souweine, Jean-Etienne Bazin, Jennifer Brunet, Jean-Baptiste Michot, Aurélien Seemann, Fabienne Tamion, Francis Remerand, Pierre-François Dequin, Adel Maamar, Yoann Launey, Pierre-Thierry Piechaud, Aurélien Frerou, Guillaume Schnell, Pierre Fillatre, Chloé Thill, Guillaume Grillet, Pierre-Yves Egreteau, Johann Auchabie, Pierre-Louis Declercq, Benoît Roze, Marc-Olivier Fischer, Loïc Dopeux, Ramin Ravan, Luc Jarrige, Christophe Faisy, Bertrand Sauneuf, Nicolas Pichon, Fabrice Prevost, Agathe Delbove, Maud Jonas, Sandrine Bedon Carte, Gaël Pradel, Antoine Vieillard-Baron, Salah Boussen, Marc Gainnier, Samuel Lehingue, Marie-Christine Herault, Michel Durand, Philippe Deswardt, Philippe Camarasa, Carole Ichai, Jean-Emmanuel Alphonsine, Bruno Marquer, Vincent Willems, Alexandre Robert, Pierre-Marie Bertrand, Nadiejda Antier, Toufic Finge, Lucie Mouton, Touraj Rastegar, Tobias Graf, Jean Reuter, Claude Braun, Stephan Watremez, Uwe Spetzger, André Michel, Marc Bodenstein.

## Author contributions

**Conceptualization:** Chatpimuk Thipayamaskomon, Olivier Grimaud, Pierre Tattevin, Lionel Lamhaut, Emmanuelle Leray, Noemie Letellier, Sahar Bayat, Christophe Fermanian, Sylvie Martin, Jean-Marc Philippe, Eric Maury, Marc Noizet, François Braun, Manuel Dolz, Marc-Antoine Sanchez, Hélène Coignard-Biehler, Nathalie Prieto, Hugues Delamare, Virginie Cayré, Pierre Carli, Albert Vuagnat, Julien Pottecher, Agnès Ricard-Hibon.

**Data curation:** Chatpimuk Thipayamaskomon, Christophe Fermanian, Sylvie Martin.

**Formal analysis:** Chatpimuk Thipayamaskomon, Christophe Fermanian, Sylvie Martin.

**Funding acquisition:** Olivier Grimaud.

**Investigation:** Olivier Grimaud, Pierre Tattevin, Lionel Lamhaut, Christophe Fermanian, Jean-Marc Philippe, Eric Maury, Marc Noizet, François Braun, Manuel Dolz, Marc-Antoine Sanchez, Hélène Coignard-Biehler, Nathalie Prieto, Hugues Delamare, Virginie Cayré, Pierre Carli, Albert Vuagnat, Julien Pottecher, Agnès Ricard-Hibon.

**Methodology:** Olivier Grimaud, Emmanuelle Leray, Sahar Bayat.

**Project administration:** Olivier Grimaud.

**Resources:** Olivier Grimaud.

**Supervision:** Olivier Grimaud, Emmanuelle Leray, Noemie Letellier.

**Validation:** Olivier Grimaud, Emmanuelle Leray, Noemie Letellier.

**Visualization:** Chatpimuk Thipayamaskomon, Olivier Grimaud, Sylvie Martin.

**Writing – original draft:** Chatpimuk Thipayamaskomon, Olivier Grimaud.

**Writing – review & editing:** Chatpimuk Thipayamaskomon, Olivier Grimaud, Pierre Tattevin, Lionel Lamhaut, Emmanuelle Leray, Noemie Letellier, Sahar Bayat, Christophe Fermanian, Sylvie Martin, Jean-Marc Philippe, Eric Maury, Marc Noizet, François Braun, Manuel Dolz, Marc-Antoine Sanchez, Hélène Coignard-Biehler, Nathalie Prieto, Hugues Delamare, Virginie Cayré, Pierre Carli, Albert Vuagnat, Julien Pottecher, Agnès Ricard-Hibon.

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
