## [Decision Letter · Decision Letter 0]

8 Feb 2026

Dear Dr. Grimaud,

Thank you for submitting your manuscript to PLOS ONE. After careful consideration, I feel that it has merit but does not fully meet PLOS ONE’s publication criteria as it currently stands. Therefore, I invite you to submit a revised version of the manuscript that addresses the points raised during the review process.

We look forward to receiving your revised manuscript.

Kind regards,

Amir Nutman

Academic Editor

PLOS One

Journal Requirements:

For additional information about PLOS ONE ethical requirements for human subjects research, please refer to http://journals.plos.org/plosone/s/submission-guidelines#loc-human-subjects-research....

4. In the online submission form you indicate that your data is not available for proprietary reasons and have provided a contact point for accessing this data. Please note that your current contact point is a co-author on this manuscript. According to our Data Policy, the contact point must not be an author on the manuscript and must be an institutional contact, ideally not an individual. Please revise your data statement to a non-author institutional point of contact, such as a data access or ethics committee, and send this to us via return email. Please also include contact information for the third party organization, and please include the full citation of where the data can be found.

5. One of the noted authors is a group or consortium “TRANSCOV Investigators”. In addition to naming the author group, please list the individual authors and affiliations within this group in the acknowledgments section of your manuscript. Please also indicate clearly a lead author for this group along with a contact email address.

6. We note that there is identifying data in the Supporting Information file  “S2-Appendix.xlsx”. Due to the inclusion of these potentially identifying data, we have removed this file from your file inventory. Prior to sharing human research participant data, authors should consult with an ethics committee to ensure data are shared in accordance with participant consent and all applicable local laws.

-Location data

Additional Editor Comments:

In addition to the reviewer's comments, I would like to highlight one editorial point for your revision. Given the context of mass transfers of critically ill patients during an infectious disease outbreak, the manuscript would benefit from a more explicit discussion of infection prevention (IPC) aspects. Specifically, please consider addressing issues such as transmission risk during transport, mitigation strategies implemented in each transport modality, and the implications of your findings for IPC planning in future large-scale transfers.(IPC) aspects. Specifically, please consider addressing issues such as transmission risk during transport, mitigation strategies implemented in each transport modality, and the implications of your findings for IPC planning in future large-scale transfers.(IPC) aspects. Specifically, please consider addressing issues such as transmission risk during transport, mitigation strategies implemented in each transport modality, and the implications of your findings for IPC planning in future large-scale transfers.(IPC) aspects. Specifically, please consider addressing issues such as transmission risk during transport, mitigation strategies implemented in each transport modality, and the implications of your findings for IPC planning in future large-scale transfers.

I look forward to receiving your revised submission.

Reviewers' comments:

Reviewer's Responses to Questions

**Comments to the Author**

1. Is the manuscript technically sound, and do the data support the conclusions?

Reviewer #1: Partly

2. Has the statistical analysis been performed appropriately and rigorously?

Reviewer #1: No

3. Have the authors made all data underlying the findings in their manuscript fully available?

Reviewer #1: No

4. Is the manuscript presented in an intelligible fashion and written in standard English?

Reviewer #1: Yes

Reviewer #1: Thank you for the opportunity to review this interesting manuscript.

I have two main reservations:

1. Length of stay in the destination ITU is used as an outcome measure to compare Train vs Air transport. However, the destinations between the two groups differ significantly (Table 2) and importantly. This is a key confounder and the most likely reason for the findings with regards to a shorter length of stay in patients transferred by train. Since they arrived in ICUs further away from the epicentre of the pandemic with likely more capacity and less requirement to discharge patients rapidly from ITU. The authors mention this but do not highlight it sufficiently (I disagree that "physical & care environments during transfer provide a more compelling explanation" for a median LOS of 6 days less in the train group). The manuscript should be adapted to make this key limitation more explicit. A geographical map summarising destinations for Train group vs Air group may be helpful to the reader. Average length of stay in ITU for the period in question at each destination Hospital would be very useful to assess the likely impact of destination ITU but appreciate this data may not be available.

2. 208 patients have been excluded from the initial 501 transfers. 171 due to unknown destination and 20 for unknown vectors. The authors also note "high proportion of missing data for some key parameters". The percentage of missing data for each parameter needs to be provided as a supplementary table - I am concerned that the SOFA score comparison presented (lines 207 to 210) may be misleading if a large volume of data points are missing. I am generally concerned with the large percentage of missing data and conclusions must be suitably cautious. The manuscript must make it explicit to the reader as to the uncertainty of any conclusions drawn from this data.

Other points for consideration

Line 82 - please include a comparison of what was available on trains in terms of procedures and tests vs air.

Paragraph from Line 85 to 90 should be in the methods section.

Line 89 "noradrenaline <1mg/h" - was this not mcg/kg/min? mg/h is not comparable between patients.

Line 110-111 helicopter vs fixed-wing are very different platforms. Even if data is incomplete would be useful to know the breakdown in the known data.

Line 135 "Catecholamine requirement" is used throughout the manuscript, does this exclude vasopressin? - would vasopressor/inotrope be preferable terminology?

Line 206-207 Only the air group is presented where as there is also a deterioration in the train group. Why is this not included? I would also ask for with-in group statistical analysis i.e paO2/Fio2 ratio before vs after in the train group & PaO2/Fio2 ratio before vs after in the air group as an assessment of statistically significant deterioration or not.

Line 284 "they had better nursing care" - this may be true but by what measure? At the very least this should be qualified with "may" - it should also be balanced with the more likely factor that they arrived in less pressured ICUs who were able to provide days of better ICU care thereby reducing the risk of nosocomial resp infections as an alternative (to my mind more likely) explanation for the findings.

Line 304-305 a cost analysis is performed in the conclusion. This should be moved to the discussion. Furthermore, I personally think this should be removed as a gross over-simplification of results based on generally poor data & extrapolating a cost saving is mis-leading.

Is there any information on the land transfer from hospital to airfield and airfield to hospital as this may be significant in some regions.

I thank the authors again for an interesting article with potential implications for any future pandemics/major incidents.

.

Reviewer #1: **Yes:** Nick HaslamNick HaslamNick HaslamNick Haslam

---

## [Author Response · Author response to Decision Letter 1]

26 Mar 2026

All answers are provided in the Response to reviewers document

---

## [Decision Letter · Decision Letter 1]

13 Apr 2026

High-speed trains versus air transport vectors for mass transfers of critically ill patients: the TRANSCOV cohort study

PONE-D-25-62199R1

Dear Dr. Grimaud,

We’re pleased to inform you that your manuscript has been judged scientifically suitable for publication and will be formally accepted for publication once it meets all outstanding technical requirements.

Kind regards,

Amir Nutman

Academic Editor

PLOS One

Additional Editor Comments (optional):

Reviewers' comments:

Reviewer's Responses to Questions

**Comments to the Author**

Reviewer #1: All comments have been addressed

2. Is the manuscript technically sound, and do the data support the conclusions?

Reviewer #1: Partly

3. Has the statistical analysis been performed appropriately and rigorously?

Reviewer #1: Yes

4. Have the authors made all data underlying the findings in their manuscript fully available?

Reviewer #1: Yes

5. Is the manuscript presented in an intelligible fashion and written in standard English?

Reviewer #1: Yes

Reviewer #1: I would like to thank and congratulate the authors for addressing all previous comments and suggestions comprehensively. This is an important publication, highlighting an alternative transport method that is rarely used. The conclusions are now in line with the evidence presented with adequate exploration of key confounders. I think this will be of interest internationally to those involved in critical care transfer and emergency planning & response.

.

Reviewer #1: **Yes:** Nick HaslamNick HaslamNick HaslamNick Haslam

---

## [Editor Report · Acceptance letter]

PONE-D-25-62199R1

PLOS One

Dear Dr. Grimaud,

I'm pleased to inform you that your manuscript has been deemed suitable for publication in PLOS One. Congratulations! Your manuscript is now being handed over to our production team.

Kind regards,

on behalf of

Dr. Amir Nutman

Academic Editor

PLOS One